# Effects of intraoperative hypothermia on patients undergoing laparoscopic surgery: A retrospective cohort study

Tashi Penjore[ID], Maliwan Oofuvong[ID]*, Sunisa Chatmongkolchart, Chanatthee Kitisiripant[ID], Rongrung Rueangchira-urai[☯], Jaranya Leeratiwong[☯]

Department of Anesthesiology, Faculty of Medicine, Prince of Songkla University, Hat Yai, Thailand

☯ These authors contributed equally to this work.
* oomaliwa@gmail.com

**Data Availability Statement:** All relevant data are within the paper and its Supporting Information files.

## Abstract

### Objective

This study aims to explore the association between intraoperative hypothermia and outcomes in adult patients undergoing laparoscopic surgery.

### Methods

A retrospective analysis of 2048 adult laparoscopic surgery patients treated between 2020 and 2021 was conducted at Songklanagarind Hospital, Thailand. Intraoperative hypothermia, defined as a core temperature below 36˚C, was recorded as either one or more than one episode. Patient demographics, clinical information and postoperative outcomes were extracted from the hospital information system. The outcomes were intraoperative and postoperative cardiac arrhythmias, postoperative oxygen requirement, length of post-anesthetic care unit (PACU) stay, and length of hospital stay. Univariate and multivariate logistic/linear regression models were fit to assess the association between hypothermia and outcomes, presented as odds ratio (OR) or beta-coefficient (β) with 95% confidence interval (CI).

### Results

The incidence of intraoperative hypothermia was 34.9%, with 688 experiencing one episode and 27 experiencing multiple episodes. On multivariate analysis, patients with one and more than one episode of hypothermia had an increased risk of intraoperative cardiac arrhythmia compared to normothermic patients (OR [95%CI]: 1.67[1.24,2.25] and 3.65 [1.53,8.74], respectively, P<0.001). Normothermic patients and hypothermic patients with more than one-episode had a higher odds of postoperative oxygen requirement compared to hypothermic patients with only one episode (OR [95%CI]:1.32[1.02,1.7] and 2.64 [1.1,6.32], respectively, P = 0.019). Hypothermic patients with one and more than one-episode also had longer PACU stays compared to normothermic patients (β[95%CI]:3.82 [1.34,6.29] and 12.43[2.29,22.57] minutes, respectively, P = 0.001). No significant differences were observed in the other outcomes.

**Funding:** The funding was granted for Dr. Tashi Penjore, amount of 44,600 baht (#65-226-8-9) by the Faculty of Medicine, Prince of Songkla University. The funder had no role in study design, data collection and analysis, decision to publish, or preparation of the manuscript.

**Competing interests:** The authors have declared that no competing interests exist.

## Conclusions

Intraoperative hypothermia in laparoscopic surgery is associated with a higher likelihood of intraoperative cardiac arrhythmias, prolonged PACU stay and higher postoperative oxygen requirement. Further research and prospective studies are warranted to validate these results and develop strategies to manage intraoperative hypothermia effectively.

## Introduction

The effect of anesthesia impairing thermoregulatory control and cold environments inside the operating theater can cause surgical patients to become hypothermic with incidences ranging from 45% to 70% [1, 2]. Inadvertent intraoperative hypothermia is usually defined as a temperature below 36˚C [3, 4]. Numerous studies have identified risk factors of hypothermia, including age, body mass index (BMI), surgery time, intraoperative irrigation, intraoperative infusion, blood transfusion and operating room temperature [3, 5]. Hypothermia in open surgery can cause surgical site infections [4, 6–8], coagulopathy and increased transfusion requirement [7–9], increased requirement of analgesics, altered drug metabolism and adverse cardiac events [4, 8]. In general, hypothermia is mostly associated with open surgery due to large visceral exposure leading to heat and moisture loss, but it can also occur in laparoscopic surgery. The incidence of hypothermia during laparoscopic surgery ranges from 29% to 90% [10, 11] and the risk factors are male [12], longer duration of anesthesia [13–15], use of dry $CO_2$ for insufflation [13–15], basal body temperature $\leq 36.1$˚C [16], operating room temperature $\leq 22$˚C [16] and volume of irrigation fluids [10]. Nevertheless, the literature on the consequences of hypothermia during laparoscopic surgery is limited with only a few studies addressing this issue and having inconsistent results [17–19]. Given these circumstances, our study specifically examined the consequences of intraoperative hypothermia during the laparoscopic surgery.

## Materials and methods

This study was approved by the Institutional Ethics Committee of the Faculty of Medicine, Prince of Songkla University, Songkhla, Thailand on 11 July 2022 (REC 65-226-8-9). Since this is a retrospective study, informed consent was waived by the Institutional Ethics Committee of the Faculty of Medicine, Prince of Songkla University. The data were accessed on 12 July 2022 after the Ethics Committee's approval. All data were fully anonymized before being accessed by the investigators. All patient, anesthesia, and surgery-related information were retrieved from the anesthetic recording system and Hospital Information System (HIS) of Songklanagarind hospital.

The data collection was performed solely by the principal researcher (TP) and research assistants (RR and JL). The information was treated as confidential, known only to the research team. The DOI link by Protocols.io is dx.doi.org/10.17504/protocols.io.14egn6d2yl5d/v1.

### Study population and sampling

This was a single center retrospective cohort study and included all adult patients who underwent laparoscopic surgery at Songklanagarind Hospital, Prince of Songkla University from January 2019 to December 2021. The inclusion criteria were patients aged >18 years who underwent laparoscopic surgery and an American society of anesthesia (ASA) classification

I-III. Cases were excluded if they had open surgery, if their temperature was not monitored, if they experienced intraoperative hyperthermia (defined as core temperature >37.8˚C) or if they were intubated and on a ventilator prior to surgery.

## Standard operating procedure

All study patients had surgery in the regular operating rooms and were provided with surgical drapes and blankets. All patients received general anesthesia. The induction agents were either propofol or thiopental depending on anesthesiologist staff. Fentanyl or morphine was the narcotics used the most. Cisatracurium or rocuronium was the neuromuscular blocking agent of choice. Tracheal intubation was done in all the cases and anesthesia was maintained with volatile anesthetic agents, narcotics, and neuromuscular blocking agents as per the choice of the anesthesiologist staff. All patients were provided with a warming mattress and forced air warming system before the start of surgery. Temperatures were measured by nurse anesthetists using various routes and were recorded in the patients chart every 30 minutes (S1 Fig). Active interventions such as force air warmer with further transfusion line warmers, along with other measures, were used if patients experienced a drop in temperature.

## Data collection

All necessary data including patient-related factors such as age, gender, body mass index (BMI), ASA classification, and surgery-related factors such as type (general or gynecologic), duration, and classification (emergency or elective), were collected from the HIS. Anesthesia-related factors including the technique and the type of agents used intraoperatively, were also recorded from the HIS. The lowest temperature measured anytime during the intraoperative period was considered for measuring hypothermia. The core body temperature was measured using the esophageal, nasopharynx, tympanic membrane, or rectal temperature. Since temperatures measured in the ear have been shown to be less accurate compared to rectal measurements, with a potential error of less than 0.5˚C [11], we adjusted the recorded temperatures by adding 0.5˚C to correct for this discrepancy.

## Variables of the study

**Dependent and independent variables for consequences of hypothermia.** Independent variables and potential confounders included patient characteristics: age (years), sex, weight (in kg), height (in meters), BMI (kg/m$^2$), ASA classification (I-III), surgery-related variables and technique of anesthesia. The main exposure variable was intraoperative hypothermia, defined as the core temperature <36˚C. The number of episodes of hypothermia during the intraoperative period were classified as either one or more than one. The severity of intraoperative hypothermia in our setting was classified as mild hypothermia (34 to <36˚C), moderate hypothermia (32 to <34˚C), and severe hypothermia (<32˚C). The dependent variables were intraoperative and postoperative cardiac arrhythmias, postoperative oxygen requirement, post-anesthetic care unit (PACU) admission, length of PACU stay, and length of hospital stay.

**Definition.** Site of operation included gastrointestinal, gynecological, genito-urinary surgery and others which included transabdominal preperitoneal repair hernia, adrenalectomy, hepatectomy, pancreatectomy and splenectomy. Duration of surgery was defined as the time from induction of anesthesia to extubation (in minutes). Cardiac arrhythmias were defined as any abnormality from the normal sinus rhythm or normal rate including bradycardia, supraventricular tachycardia, and atrial fibrillation. Postoperative oxygen requirement was noted if the patient had utilized oxygen in the PACU or at the ward. Delayed awakening was defined if more than 20–30 minutes elapsed before the patient woke up after the end of anesthesia.

Intraoperative or postoperative blood transfusion (ml) was defined if a patient with hypothermia received blood transfusion either during the operation or after the operation. Length of postoperative hospital stay (in days) was defined as the time from surgery until discharge from hospital. Duration of hypothermia was defined as the period during which core body temperature was <36˚C measured in 30 minute intervals. If within any 30 minute period, the body temperature fluctuated from normal to below 36˚C or vice versa, 15 minutes was added to the total duration. Length of PACU stay (in minutes) was defined from the time from finishing the operation until discharge from recovery room.

**Statistical analysis.** Data were entered with Epidata (version 3.1). All variables were presented descriptively with mean and standard deviation or median and interquartile range as appropriate for continuous variables, and frequency and percent for categorical variables. Unpaired Student's t-tests or Wilcoxon's rank sum tests were used to compare normally or non-normally distributed variables, respectively. The chi-square test and Fisher's exact test were used to compare categorical variables. Logistic regression and linear regression were used to determine the association between intraoperative hypothermia and the outcomes depending on the type of outcome. Variables with a P-value <0.1 from the univariate analyses were included in the multivariate regression models. We assessed the distribution of residuals from the linear model and, if it was not normal, we transformed the outcome by taking its logarithm. We also assessed the model for multicollinearity and homoscedasticity to satisfy linear assumptions. The model was then refined by sequential backward elimination of non-significant variables performed by the likelihood ratio test, providing the odds ratios (OR)/beta coefficient (β) and their 95% confidence intervals. We subsequently removed each variable with the highest P value and assessed the AIC value for that model. The model with the lowest AIC value as accepted as our final model. Factors were considered significant if their likelihood ratio test P values were < 0.05. The missing data were reported and automatically excluded in the analysis process.

## Sample size calculations

The sample size calculation was performed based on a pilot study by reviewing the anesthetic records of laparoscopic surgery cases between January and December 2021 (115 hypothermia, 980 non-hypothermia) to achieve the primary objective under a power of 80% and a level of significance of 0.05. Based on the sample size formula for comparing two proportions, at least 2000 patients were required.

## Results

Data were collected from 2200 patients undergoing laparoscopic surgery between January 2019 and December 2022. We excluded 152 patients: 36 had open surgery, 74 did not have intraoperative temperature monitored, 33 had intraoperative hyperthermia, and 9 were ASA classification IV. Among the remaining 2,048 patients (S1 File), 714 (34.9%) experienced intraoperative hypothermia (Fig 1). The characteristics of the study participants are presented in Table 1.

From the total cases undergoing laparoscopic surgery, 733 (35.8%) were male and 1,315 (64.2%) were female. Among the 715 patients who experienced hypothermia, 387 (54.1%) were male and 328 (45.9%) were female. Patients with hypothermia was significantly older. The majority of the patients were classified as ASA class II. All other baseline characteristics were also statistically significant. The route of temperature monitoring was not different between the two groups. A comparison of outcomes between patients with normothermia and hypothermia are shown in S1 Table. Among the 715 hypothermic patients, 688 had only 1

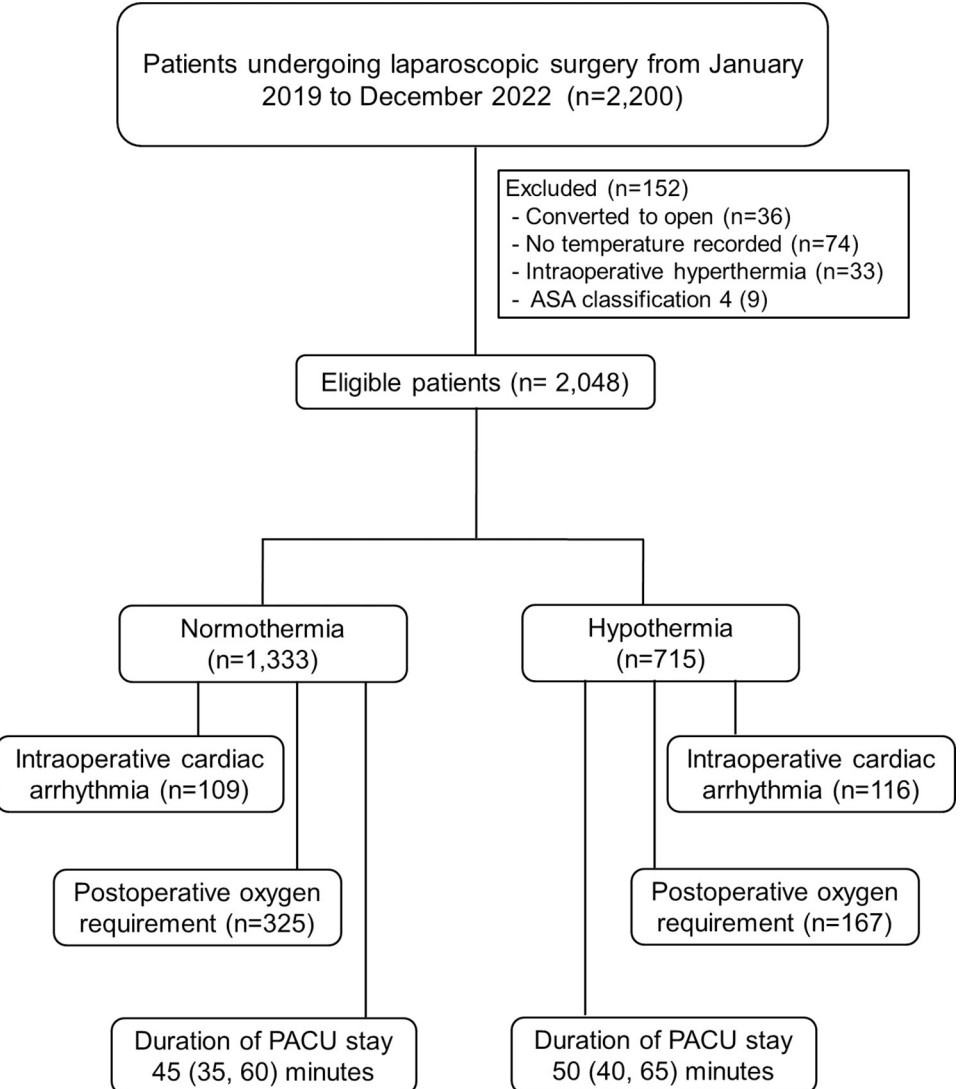

**Fig 1. Flow diagram of the study.** PACU = Post-anesthetic care unit.

episode and 27 had more than 1 episode (Table 2). The outcomes among episodes of hypothermia are shown in Table 3.

Among the 1,333 normothermic patients, 109 (8.2%) experienced cardiac arrhythmias while among the 688 and 27 patients with one and more than one episode of hypothermia, 108 (15.7%) and 8 (29.6%) patients had cardiac arrhythmias, respectively. In normothermic patients, 33 (2.5%) required intraoperative blood transfusion, while patients with one and more than one episode of hypothermia, 23 (3.3%) and 3 (11.1%) patients, respectively, required intraoperative blood transfusion. Normothermic patients stayed in the PACU for an average of 45 minutes, whereas patients who experienced one and more than one episode of hypothermia had an average stay of 50 and 55 minutes, respectively. We found that 44.4% of patients with more than one episode of hypothermia required postoperative oxygen compared to 24.4% for normothermic patients and 22.4% for patients who experienced only one episode. Patients with more than one episode of hypothermia had a longer duration of hospital stay than the other two groups. There were no other significant differences between the three

**Table 1. Baseline characteristic of the participants.**

| Variables | Hypothermia | | P value |
|---|---|---|---|
| | No (N = 1333) | Yes (N = 715) | |
| Male | 346 (26) | 387 (54.1) | < 0.001 |
| **Age (years),** median (IQR) | 47 (37,63) | 61 (49,70) | < 0.001 |
| **Age group (year)** | | | < 0.001 |
| 18–49 | 729 (54.7) | 182 (25.5) | |
| 50–65 | 332 (24.9) | 253 (35.4) | |
| >65 | 272 (20.4) | 280 (39.2) | |
| **Body mass index (kg/m$^2$),** median (IQR) | 24.9 (22,29.5) | 24 (21.4,27) | < 0.001 |
| **ASA Classification** | | | 0.002 |
| I | 114 (8.6) | 34 (4.8) | |
| II | 841 (63.1) | 494 (69.1) | |
| III | 378 (28.4) | 187 (26.2) | |
| **Preoperative baseline temperature,** median (IQR) | 36.7 (36.6,36.9) | 36.6 (36.5,36.8) | < 0.001 |
| **Site of Surgery** | | | < 0.001 |
| Gastrointestinal | 775 (58.1) | 507 (70.9) | |
| Gynecology | 445 (33.4) | 99 (13.8) | |
| Genitourinary | 80 (6) | 84 (11.7) | |
| Others (adrenalectomy, hepatectomy, splenectomy pancreatectomy) | 33 (2.5) | 25 (3.5) | |
| **Timing** | | | < 0.001 |
| Elective | 1277 (95.8) | 706 (98.7) | |
| Emergency | 56 (4.2) | 9 (1.3) | |
| **Anesthesia duration** (minutes), median (IQR) | 170 (115,260) | 200 (135,330) | < 0.001 |
| **Underlying cardiovascular disease** | | | < 0.001 |
| Hypertension | 440 (33) | 285 (39.9) | |
| Other (heart failure, ischemic heart disease, arrhythmias) | 155 (11.6) | 109 (15.2) | |
| **Underlying respiratory disease** | | | < 0.001 |
| Asthma | 44 (3.3) | 18 (2.5) | |
| Obstructive sleep apnea | 147 (11) | 47 (6.6) | |
| Others (pneumonia, allergic rhinitis, tuberculosis, COPD) | 65 (4.9) | 59 (8.3) | |
| **Underlying neurological disease** | | | 0.003 |
| Cerebrovascular accident | 42 (3.2) | 31 (4.3) | |
| Other (Parkinson's, epilepsy, muscular dystrophy) | 28 (2.1) | 32 (4.5) | |
| **Underlying hematologic disease** | | | 0.031 |
| Anemia | 391 (29.3) | 250 (35) | |
| Coagulopathy | 5 (0.4) | 3 (0.4) | |
| Underlying malignancy | 319 (23.9) | 274 (38.3) | < 0.001 |
| Underlying diabetes mellitus: | 211 (15.8) | 108 (15.1) | 0.714 |
| Underlying hypothyroidism | 17 (1.3) | 11 (1.5) | 0.772 |
| Underlying hyperthyroidism | 11 (0.8) | 7 (1) | 0.915 |
| Underlying dyslipidemia | 418 (31.4) | 273 (38.2) | 0.002 |
| Estimated blood loss median (IQR) | 20 (10,50) | 20 (10,100) | 0.005 |
| **Route of temperature monitoring** | | | 0.334 |
| Nasopharynx | 1137 (85.3) | 620 (86.7) | |
| Esophagus | 83 (6.2) | 50 (7) | |
| Tympanic membrane | 110 (8.2) | 44 (6.2) | |

(*Continued*)

**Table 1.** (Continued)

| Variables | Hypothermia | | P value |
|---|---|---|---|
| | **No (N = 1333)** | **Yes (N = 715)** | |
| Rectum/axilla | 3 (0.2) | 1 (0.1) | |

Notes: Data are presented as frequency (%) or median (interquartile range [IQR]) unless stated otherwise. ASA = American Society of Anesthesiologists, COPD = Chronic obstructive lung disease.

groups in terms of the other outcomes on univariate analysis. Even though the hypothermia group was related to a higher proportion of blood transfusion requirements, it was not statistically significant in the multivariate logistic regression model.

## Intraoperative cardiac arrhythmia

From the multivariate logistic regression analysis shown in Table 4, hypothermic patients were significantly more likely to develop intraoperative cardiac arrhythmias after adjusting for age and underlying disease. Hypothermic patients with only one episode were 1.65 times more likely to experience intraoperative cardiac arrhythmias compared to normothermic patients (OR (95%CI);1.67(1.24,2.25), P<0.001) and patients with more than one episode of hypothermia were three times as likely (OR (95%CI);3.65(1.53,8.74), P<0.001). Receiver operating characteristic (ROC) curve analysis was performed which demonstrated an area under ROC curve of 0.669 (S2 Fig), indicating a fair to moderate discriminatory power in predicting intraoperative cardiac arrhythmias based on presence and absence of intraoperative hypothermia.

## Postoperative oxygen requirement

Postoperative oxygen requirement was significantly associated with hypothermia (P = 0.029) as shown in Table 5.

Patients experiencing more than one episode of hypothermia were more likely to have post-operative oxygen requirement (OR (95% CI); 2.64 (1.1,6.32), P = 0.019) compared to those with normothermia and one episode of hypothermia. Additionally, patients with normothermia were also more likely to require postoperative oxygen compared to those with only

**Table 2.** Characteristics of patients with hypothermia (N = 715).

| Characteristic | Hypothermia |
|---|---|
| **Nadir temperature (˚C)** | |
| Mean (SD) | 35.6 (0.29) |
| Median (IQR) | 35.7 (35.5, 35.8) |
| **Duration of hypothermia (minutes)** | |
| Mean (SD) | 80.5 (58.9) |
| Median (IQR) | 60 (45, 105) |
| **Episode of hypothermia** | |
| 1 | 688 (96.2) |
| 2 | 24 (3.4) |
| >2 | 3 (0.4) |
| **Severity of hypothermia** | |
| Mild (34 to <36˚C) | 715 (100%) |

Note: Data are presented as frequency (%) unless stated otherwise. *IQR* Interquartile range, *SD* Standard deviation.

**Table 3. Comparison of outcomes between patients with normothermia, one episode of hypothermia and more than one episode of hypothermia.**

| Outcome variable | Normothermia (N = 1333) | 1 episode of hypothermia (N = 688) | >1 episode of hypothermia (N = 27) | P value |
|---|---|---|---|---|
| **Intraoperative cardiac arrhythmias** | 109 (8.2) | 108 (15.7) | 8 (29.6) | <0.001 |
| **Intraoperative blood transfusion** | 33 (2.5) | 23 (3.3) | 3 (11.1) | 0.02 |
| **Volume of intraoperative blood transfusion (ml),** median (IQR) | 0 (0,0) | 0 (0,0) | 0 (0,0) | 0.793 |
| Mean (SD) | 11.73 (91.12) | 15.45 (115.1) | 29.93 (155.1) | 0.462 |
| **PACU stay** | 1323 (99.2) | 675 (98.1) | 25 (92.6) | 0.001 |
| **Duration of PACU stay (minutes),** median (IQR) | 45 (35,60) | 50 (40,65) | 55 (45,75) | <0.001 |
| **Delayed awakening** | 3 (0.2) | 4 (0.6) | 0 (0) | 0.41 |
| **Shivering** | 29 (2.2) | 17 (2.5) | 0 (0) | 0.667 |
| **Nausea/vomiting** | 151 (11.3) | 72 (10.5) | 2 (7.4) | 0.703 |
| **Postoperative oxygen requirement** | 325 (24.4) | 155 (22.5) | 12 (44.4) | 0.029 |
| **Duration of oxygen requirement (hrs),** median (IQR) | 15 (12,18) | 16 (12,18) | 18.5 (12.8,24) | 0.242 |
| **Type of ventilator** | | | | <0.001 |
| Endotracheal intubation | 12 (0.9) | 14 (2) | 3 (11.1) | |
| NPPV | 25 (1.9) | 4 (0.6) | 1 (3.7) | |
| **Postoperative ICU** | 13 (1) | 11 (1.6) | 5 (18.5) | <0.001 |
| **Duration of ICU stay (days),** median (IQR) | 1 (1,2) | 3 (2.5,4.5) | 3 (2,3) | 0.061 |
| **Duration of hospital stay (days),** median (IQR) | 4 (3,5) | 4 (3,6) | 7 (5.5,8) | <0.001 |
| **Postoperative cardiac arrhythmia** | 2 (0.2) | 2 (0.3) | 0 (0) | 0.773 |

**Notes:** Data are presented as frequency (%) or median (interquartile range [IQR]) unless stated otherwise. ICU = Intensive care unit, NPPV = Non-invasive positive pressure ventilation, PACU = Post-anesthetic care unit, SD = Standard deviation.

one episode of hypothermia (OR (95%CI); 1.32 (1.02,1.7), P = 0.019. The area under the ROC curve was 0.8 (S3 Fig), indicating a good discriminatory power in predicting postoperative oxygen requirement based on presence and absence of intraoperative hypothermia.

## Duration of PACU stay

Hypothermia was significantly associated with an increased duration of PACU stay. Patients with one or more than one episode had longer PACU stays compared to normothermic patients ($\beta$[95%CI]:3.82[1.34,6.29] and 12.43[2.29,22.57] minutes, respectively, P = 0.001)

**Table 4. Final model of multivariate analysis for intraoperative cardiac arrhythmias.**

| Variables | Crude OR (95% CI) | Adjusted OR (95% CI) | P (Wald's test) | P (LR-test) |
|---|---|---|---|---|
| **Hypothermia episodes** (ref. = normothermia) | | | | < 0.001 |
| 1 | 2.09 (1.57,2.78) | 1.67 (1.24,2.25) | < 0.001 | |
| >1 | 4.73 (2.02,11.05) | 3.65 (1.53,8.74) | 0.004 | |
| **Age group, years** (ref. = 18–49) | | | | < 0.001 |
| 50–65 | 2.12 (1.48,3.02) | 1.76 (1.21,2.56) | 0.003 | |
| >65 | 2.73 (1.93,3.86) | 2.08 (1.39,3.1) | < 0.001 | |
| **Underlying cardiovascular disease** (ref. = none) | | | | 0.013 |
| Hypertension | 1.73 (1.27,2.35) | 1.46 (1.03,2.08) | 0.035 | |
| Others (heart failure, ischemic heart disease, arrhythmias) | 2.2 (1.48,3.26) | 1.79 (1.19,2.69) | 0.005 | |
| **Hyperparathyroidism or hyperaldosteronism** | 0.58 (0.38,0.89) | 0.47 (0.3,0.73) | < 0.001 | < 0.001 |

**Notes:** LR-test = likelihood ratio test. OR = odds ratio, CI = confidence interval.

**Table 5. Final model of multivariate analysis for postoperative oxygen requirement.**

| Variables | Crude OR (95% CI) | Adjusted OR (95% CI) | P (Wald's test) | P (LR-test) |
|---|---|---|---|---|
| **Hypothermia episodes** (ref. = 1) | | | | 0.019 |
| None | 1.11 (0.89,1.38) | 1.32 (1.02,1.7) | 0.035 | |
| >1 | 2.75 (1.26,6) | 2.64 (1.1,6.32) | 0.029 | |
| **Age group, years** (ref. = 18–49) | | | | < 0.001 |
| 50–65 | 1.03 (0.79,1.33) | 1.89 (1.33,2.7) | < 0.001 | |
| >65 | 2.2 (1.73,2.79) | 3.88 (2.67,5.63) | < 0.001 | |
| **Body mass index (kg/m²)** | 1.09 (1.07,1.1) | 1.07 (1.04,1.09) | < 0.001 | < 0.001 |
| **ASA Classification** (ref. = I) | | | | 0.002 |
| II | 2.72 (1.41,5.25) | 0.9 (0.44,1.81) | 0.758 | |
| III | 12.0 (6.17,23.2) | 1.5 (0.71,3.16) | 0.286 | |
| **Duration of anesthesia** (minutes) | 1.0028 (1.0021,1.0034) | 1.0022 (1.0015,1.003) | < 0.001 | < 0.001 |
| **Underlying cardiovascular disease** (ref. = no) | | | | 0.033 |
| Hypertension | 2.88 (2.31,3.6) | 1.46 (1.09,1.96) | 0.01 | |
| Others (heart failure, ischemic heart disease, arrhythmias) | 1.44 (1.03,2.01) | 1.13 (0.77,1.67) | 0.528 | |
| **Underlying respiratory disease** (ref. = no) | | | | < 0.001 |
| Asthma | 2.1 (1.22,3.63) | 1.46 (0.8,2.69) | 0.22 | |
| Obstructive sleep apnea | 8.89 (6.44,12.26) | 3.26 (1.98,5.35) | < 0.001 | |
| Others (pneumonia, allergic rhinitis, tuberculosis, COPD) | 1.6 (1.05,2.43) | 1.55 (0.97,2.46) | 0.066 | |
| **Underlying diabetes mellitus** | 2.68 (2.09,3.45) | 1.27 (0.94,1.72) | 0.113 | 0.114 |

**Notes:** LR-test = likelihood ratio test. OR = odds ratio, CI = confidence interval, ASA = American society of anesthesia, COPD = Chronic obstructive lung disease.

(Table 6). ASA classification, underlying respiratory disease and underlying dyslipidemia were also associated with PACU stay.

## Duration of hospital stay

From the univariate linear regression model, there was a significant association between hypothermia and duration of hospital stay. However, on multivariate analysis, the association was not statistically significant. ASA classification, duration of anesthesia, underlying malignancy, underlying hematologic conditions and the timing of surgery were significantly associated with duration of hospital stay. Fig 2 displays the associations between hypothermia and perioperative outcomes.

Normothermic patients and those with more than one episode of hypothermia had increased odds of postoperative oxygen requirement compared to patients with one episode of hypothermia (OR [95%CI]:1.32[1.02,1.7] and 2.64[1.1,6.32], respectively, P = 0.019). Patients with one and more than one episode of hypothermia had almost double and more than triple the risk of intraoperative cardiac arrhythmia compared to normothermic patients (OR [95% CI]: 1.67[1.24,2.25] and 3.65[1.53,8.74], respectively, P<0.001). Patients with one and more than one episode of hypothermia experienced increased duration of PACU stay compared to normothermic patients (β[95%CI]:3.82[1.34,6.29] and 12.43[2.29,22.57] minutes, respectively, P = 0.001). No significant differences were observed in the other outcomes.

## Discussion

This study marks one of the initial investigations examining the direct implications of intraoperative hypothermia in patients undergoing laparoscopic surgery. This retrospectively study

**Table 6. Multiple linear regression model for duration of PACU stay.**

| Variables | Crude β (95%CI) | Adjusted β (95%CI) | P (Wald's test) | P (F-test) |
|---|---|---|---|---|
| **Hypothermia episodes**: (ref. = none) | | | | 0.001 |
| 1 | 4.13 (1.75,6.51) | 3.82 (1.34,6.29) | 0.002 | |
| >1 | 15.25 (5.11,25.4) | 12.43 (2.29,22.57) | 0.013 | |
| **Age group, years** (ref. = 18–49) | | | | 0.004 |
| 50–65 | -2.68 (-5.37,0) | -4.9 (-7.94,-1.85) | 0.002 | |
| >65 | 2.24 (-0.5,4.98) | -2.67 (-6.13,0.79) | 0.13 | |
| **ASA classification** (ref. = I) | | | | < 0.001 |
| II | 3.47 (-0.79,7.74) | 0.36 (-4.09,4.81) | 0.874 | |
| III | 15.81 (11.25,20.36) | 7.79 (2.64,12.94) | 0.003 | |
| **Site of surgery** (ref. = Gynecology) | | | | 0.004 |
| Others (adrenalectomy, hepatectomy, splenectomy pancreatectomy) | -15.1 (-22.11, -8.09) | 10.59(3.61,17.56) | 0.003 | |
| Genitourinary | -5.76 (-13.51,1.98) | 5.74(0.94,10.55) | 0.019 | |
| Gastrointestinal | -7.21 (-14.03,-0.4) | 4.01(1.05,6.98) | 0.008 | |
| **Duration of anesthesia** (minutes) | 0.02 (0.01,0.03) | 0.01 (0,0.01) | 0.235 | 0.235 |
| **Underlying respiratory disease** (ref. = no) | | | | <0.001 |
| Asthma | 3.53 (-2.87,9.93) | 0.88 (-5.45,7.21) | 0.785 | |
| Obstructive sleep apnea | 17.3 (13.53,21.06) | 9.76 (5.06,14.46) | <0.001 | |
| Others (pneumonia, allergic rhinitis, tuberculosis, COPD) | 6.8 (2.12,11.48) | 5.57 (0.95,10.19) | 0.018 | |
| **Underlying malignancy** | 3.1 (0.62,5.58) | 3.01(-0.05,6.07) | 0.054 | 0.054 |
| **Underlying hyperthyroidism** | -12.84 (-24.78,-0.9) | -11.55 (-23.03,-0.07) | 0.049 | 0.049 |
| **Underlying dyslipidemia** | 5.17 (2.81,7.54) | 2.62 (0.06,5.18) | 0.045 | 0.045 |

**Notes:** ASA = American Society of Anesthesiologists, β = Beta coefficient, CI = Confidence interval, COPD = Chronic obstructive lung disease.

analysed over 2,000 adult patients undergoing laparoscopic surgery in a single university hospital between 2019 to 2021. Despite the routine employment of preventive strategies within our institution, the incidence of intraoperative hypothermia during laparoscopic surgery remained high at 34.9% which is consistent with other studies [10, 20]. Cumin et al. [20] found that there were no significant differences in the risk and incidence of intraoperative hypothermic episodes between open and laparoscopic colorectal surgery. The statistical significance of baseline characteristics in our study may be due to the large sample size and may not be clinically meaningful. The primary objectives of our study was to elucidate the effects of intraoperative hypothermia on various outcomes, such as cardiac arrhythmias, postoperative oxygen requirement, and length of PACU stay. Hypothermia was defined as a temperature below 36°C as commonly considered in clinical practice. In addition to considering temperature below 36°C as our primary exposure, we also recognized the varying frequency of hypothermia in laparoscopic surgery, possibly due to the prolonged duration of these procedures. Therefore, we stratified our exposure into single and multiple episodes of hypothermia to investigate potential associations. Our findings revealed significant associations between hypothermia and outcomes such as intraoperative cardiac arrhythmias, longer PACU stay, and postoperative oxygen requirements. Initially, there was no significant association between postoperative oxygen requirement and hypothermia compared to normothermia. However, subgroup analysis showed a significant association with increased frequency of hypothermia.

Our findings suggested that a higher frequency of intraoperative hypothermic episodes correlated with a greater likelihood of patients experiencing intraoperative cardiac arrhythmias.

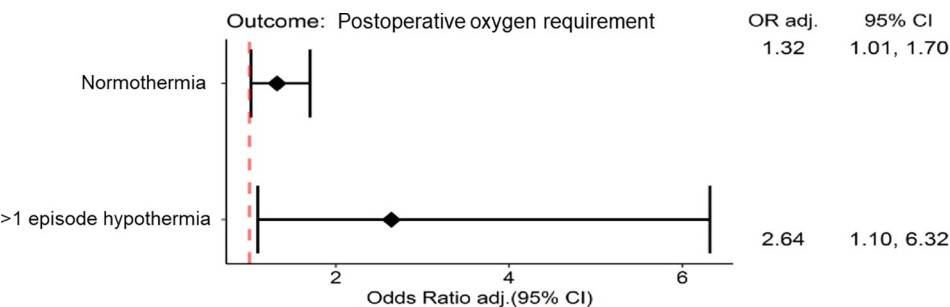

**Reference: 1 episode of hypothermia**

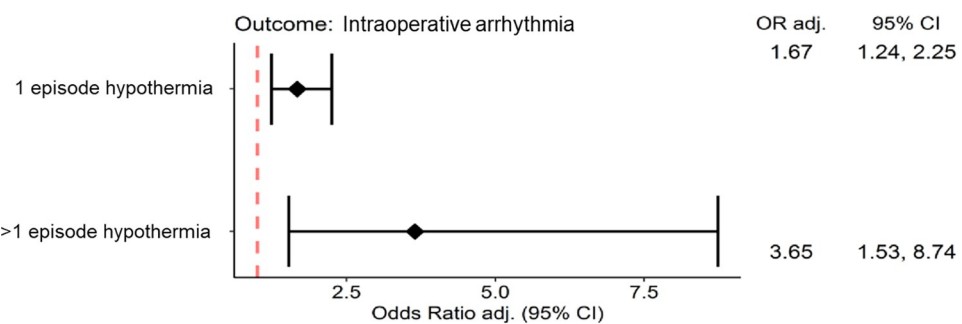

**Reference: normothermia**

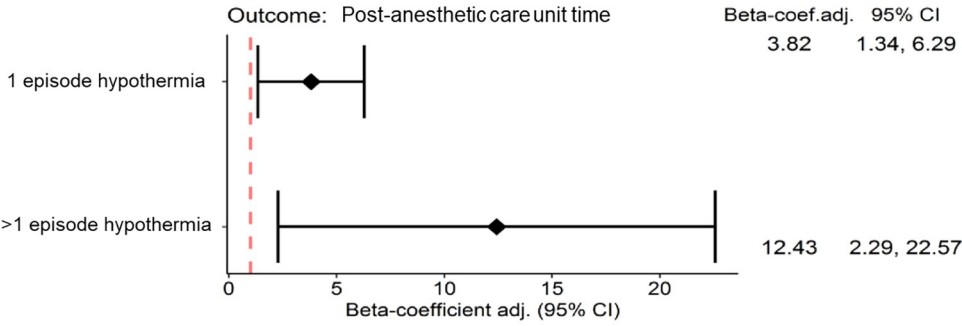

**Reference: normothermia**

**Fig 2. Association between hypothermia and perioperative outcomes by multivariate analysis.**

Specifically, individuals with one episode of hypothermia were twice as likely to experience arrhythmias, while those with two or more episodes were three times as likely. Hypothermia typically leads to a decrease in spontaneous heart rate and cardiac output as noted by Polderman [21] and Wood [22]. However, there is currently lack of consensus regarding the effects of hypothermia upon the contractile function, and the underlying physiological mechanism remains unknown.

Patients who experienced multiple hypothermic episodes exhibited higher oxygen requirement during the postoperative period. Interestingly, normothermic patients were more likely to require higher oxygen levels than those who encountered only one episode of hypothermia. This observation may be attributed to the fact that once a patient's temperature drops,

increased interventions are performed which potentially elevated their temperature by the end of surgery compared to those who remained normothermic. Hypothermia has been demonstrated to result in vasoconstriction and shivering, leading to an increase in oxygen requirements [23].

Our study revealed that patients experiencing multiple episodes of hypothermia had a prolonged duration of PACU stay compared with normothermic patients. Hypothermic patients can take a longer time to rewarm and reach acceptable temperature before discharge from PACU. A study in 2022 also suggested that intraoperative hypothermia leads to increased length of PACU stay [24].

## The association between hypothermia and other outcomes

To our knowledge, there are limited studies directly addressing the complications of intraoperative hypothermia in laparoscopic surgery. Reported consequences from few studies include increased risk of postoperative transfusion requirement [25] and increased length of stay [18, 26]. Pu et al. [27] also conducted a randomized controlled trial and found that the use of an underbody warming system can prevent hypothermia and reduce the coagulation dysfunction. Similarly, studies suggested that forced air warming can reduce the incidence of perioperative hypothermia, subsequently decreasing complications such as bleeding [28] and shivering [24, 28]. However, our study could not confirm any association between hypothermia and transfusion requirement, nor did it find any association between hypothermia and shivering. This could be due to effect of using force air warming in our setting resulting in decreasing incidence of hypothermia and its effect as seen in previous studies [29–34]. A meta-analysis conducted in 2016 on the use of warm, humidified insufflation highlighted benefits such as reduced hypothermia and shorter hospital stay [18]. Contrary to these findings, our multivariate analysis failed to find a significant association between intraoperative hypothermia and length of hospital stay. Notably, variables such as ASA classification, duration of anesthesia, underlying malignancy, underlying hematologic conditions and the timing of surgery showed significant associations with the duration of hospital stay.

## Implications of the study

The findings from this study have significant clinical implications for the management of intraoperative hypothermia in laparoscopic surgery. This study highlights the critical need for the increased awareness and proactive measures to mitigate hypothermia related complications. Optimized temperature management guidelines that we plan to implement into our institute for laparoscopic surgery are as follows. At the preoperative period, proper preoperative warming for at least 30 minutes should be considered in high-risk patients as found previously [25]; however, one should be aware that the risk of burns and thermal injuries may increase. At the intraoperative period, continuously monitoring and maintaining core body temperature above 36°C must be performed by various methods such as forced air warming devices applied under or above the patient, optimal operating room temperature, warm intravenous fluids, limiting the exposure time of surgical sites, and using warmed, humidified insufflation gases. In the PACU period, continuous warming strategies should still be considered if the core temperature remains below 36°C to minimize possible adverse events. All staff should be trained on management protocols and made aware of their importance and the possible complication of aggressive warming.

Effective temperature management could also contribute to shorter recovery times leading to improved resource utilization within health care facilities and therefore might have a direct impact on hospital efficiency and patient turnover.

## Strength and the limitations

A strength of this study is the large sample size, encompassing 2048 patients undergoing laparoscopic surgery within a single center. By comparing the outcomes between normothermic group and the hypothermic groups, we were able to assess the impact of hypothermia. Moreover, there were no missing data in the final outcome model. However, there are several limitations that warrant considerations. Firstly, the present study has all the inherent limitations of a retrospective study. All the data were extracted solely from the hospital information system, implying that our results relied significantly on the quality of the documentations. Secondly, even though the temperature was monitored continuously, it had only been recorded every 30 minutes; this could somehow affect the temperature's accuracy. However, the duration of hypothermia was also measured to see the trend of the temperature change to bring more accuracy to the temperature's data. Additionally, the variations in temperature measurement also needs to be addressed. The ambient temperatures inside the operating rooms were not documented, which could have influenced the internal core temperature readings. However, a recent study concluded that the ambient intraoperative temperature has a minimal impact on core temperature when patients are warmed with forced air [35]. Despite these measures, there might still be other confounders for hypothermia that remained uncontrolled. Moreover, being a single-center study, our results may not be generalizable to other populations or settings due to the potential differences in baseline characteristics and also the routine practices across different institutions. Future studies should address these limitations and employ more comprehensive methodologies to enhance the validity and generalizability of the findings.

## Conclusions

The incidence of intraoperative hypothermia in laparoscopic surgery in our setting was 34.9%. Our findings underscore the significant impact of intraoperative hypothermia as more episodes were associated with a higher likelihood of patients experiencing intraoperative cardiac arrhythmias, prolonged PACU stay and higher postoperative oxygen requirement. Our results emphasize the critical importance of maintaining normothermia through vigilant monitoring and proactive management during laparoscopic procedures to minimize negative outcomes.

Moving forward, it is imperative to conduct further researches, including prospective studies and randomized trials with active thermal interventions to identify its benefits, validate these results and develop optimal strategies for preventing and managing intraoperative hypothermia in laparoscopic surgery. Early recognition and efficient intervention protocols are essential during the occurrence of intraoperative hypothermia and ultimately improve the surgical outcomes and overall well-being of patients.

## Supporting information

**S1 Fig. Anesthetic record with temperature monitoring of laparoscopic surgery patient.**
(TIF)

**S2 Fig. Receiver operating characteristic curve of the final model for predicting intraoperative cardiac arrhythmia.**
(TIF)

**S3 Fig. Receiver operating characteristic curve of the final model for predicting postoperative oxygen requirement.**
(TIF)

**S1 Table. Outcome comparison between normothermia and hypothermia.**
(DOCX)

**S1 File. Minimal data set.**
(CSV)

## Acknowledgments

The authors express their deepest gratitude to all the faculty members of the master's in clinical research and health sciences program at Prince of Songkla University Additionally, we would like to acknowledge all staff in the operating room and the Department of Anesthesiology, Prince of Songkla University for their support during the course of this study.

## Author Contributions

**Conceptualization:** Tashi Penjore, Maliwan Oofuvong.

**Formal analysis:** Tashi Penjore, Maliwan Oofuvong.

**Investigation:** Tashi Penjore, Rongrung Rueangchira-urai, Jaranya Leeratiwong.

**Methodology:** Tashi Penjore, Maliwan Oofuvong, Sunisa Chatmongkolchart, Chanatthee Kitisiripant.

**Supervision:** Maliwan Oofuvong, Sunisa Chatmongkolchart, Chanatthee Kitisiripant.

**Validation:** Maliwan Oofuvong.

**Writing – original draft:** Tashi Penjore.

**Writing – review & editing:** Maliwan Oofuvong, Sunisa Chatmongkolchart, Chanatthee Kitisiripant, Rongrung Rueangchira-urai, Jaranya Leeratiwong.

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
