## [Decision Letter · Decision Letter 0]

2 Oct 2024

PONE-D-24-29030Effects of Intraoperative Hypothermia on Patients Undergoing Laparoscopic Surgery: A Retrospective Cohort StudyPLOS ONE

Dear Dr. Oofuvong,

Thank you for submitting your manuscript to PLOS ONE. After careful consideration, we feel that it has merit but does not fully meet PLOS ONE’s publication criteria as it currently stands. Therefore, we invite you to submit a revised version of the manuscript that addresses the points raised during the review process.

**ACADEMIC EDITOR: t**he paper is well-written and structured. In order to be evaluated for publication, changes must be made according to the reviewers' requests.

We look forward to receiving your revised manuscript.

Kind regards,

Marco Clementi, Associate Professor

Academic Editor

PLOS ONE

Journal Requirements:

2. Thank you for stating the following financial disclosure: The funding was granted for Dr. Tashi Penjore, amount of 44,600 baht (#65-226-8-9) by the Faculty of Medicine, Prince of Songkla University. 

Additional Editor Comments:

The paper is well-written and structured. In order to be evaluated for publication, changes must be made according to the reviewers' requests.

Reviewers' comments:

Reviewer's Responses to Questions

**Comments to the Author**

1. Is the manuscript technically sound, and do the data support the conclusions?

Reviewer #1: Yes

Reviewer #2: Yes

2. Has the statistical analysis been performed appropriately and rigorously? 

Reviewer #1: Yes

Reviewer #2: I Don't Know

3. Have the authors made all data underlying the findings in their manuscript fully available?

Reviewer #1: Yes

Reviewer #2: Yes

4. Is the manuscript presented in an intelligible fashion and written in standard English?

Reviewer #1: Yes

Reviewer #2: Yes

5. Review Comments to the Author

Reviewer #1: This is a well written retrospective review of patients undergoing laparoscopic surgery and experience hypothermia. The sample size is considerable and the analysis was sound.

Technically, retrospective review should not be called cohort study as those are typically prospective. The definitions otherwise are quite clear and allows repeatability of this study. The incidence of hypothermia was just over a third of the patients which is quite significant. Can the author provide reasons why did happen given it is a major element in the Surgical Care bundle and part of patient setup?

Can the author provide insights on how this can be prevented and that high percentage be brought down? Would adding it to the checklist help? How does this figures compare to international literature? It would hugely beneficial if the authors provide insights to how it happens and recommendations on how to prevent it.

Reviewer #2: This study explores the impact of intraoperative hypothermia on patients undergoing laparoscopic surgery, which is clinically innovative and of great significance. The research method is a retrospective cohort study with a large and comprehensive sample size, which is reliable. The setting of independent and dependent variables is reasonable and comprehensive, the statistical methods are reasonable, logistic regression and linear regression are used to determine the relationship, and there is also a clear flowchart. But there are also some shortcomings:

1.It is recommended to replace the reference with the latest one as it has a relatively long history.

2.The conclusion section can provide a clearer indication of the specific direction and focus of future research.

3.In table4, table5, and table6, abbreviations for some variables (such as OR, CI, etc.) should be given their full names and explanations when they first appear to facilitate readers' understanding. For all tables, it is recommended to add more statistical information, such as the specific calculation method of P-value and the confidence level of the confidence interval.

6. PLOS authors have the option to publish the peer review history of their article (what does this mean?). If published, this will include your full peer review and any attached files.

Reviewer #1: No

Reviewer #2: No

---

## [Author Response · Author response to Decision Letter 0]

8 Oct 2024

Dear PLOS ONE Editor,

We would like to resubmit a manuscript entitled “Effects of Intraoperative Hypothermia on Patients Undergoing Laparoscopic Surgery: A Retrospective Cohort Study” as an original article in your journal.

We are thankful to have a chance to revise our manuscript. To address the editor’s comment, we have added the information in the Tables and discussion section, updated the reference, and deposited our protocol in protocols.io. We also highlighted the changes in the current version. Attached is a point-by-point response to the reviewer’s concerns. 

Comments from the Editors

Response: We ensure that our manuscript meets PLOS ONE's style requirements, including those for file naming.

2. Thank you for stating the following financial disclosure: The funding was granted for Dr. Tashi Penjore, amount of 44,600 baht (#65-226-8-9) by the Faculty of Medicine, Prince of Songkla University. 

Response: The funder had no role in study design, data collection and analysis, decision to publish, or preparation of the manuscript. We include this amended Role of Funder statement in our recent cover letter.

Additional Editor Comments:

The paper is well-written and structured. In order to be evaluated for publication, changes must be made according to the reviewers' requests.

Response: Thank you.

Reviewers' comments:

Comments to the Author

Reviewer #1: 

This is a well written retrospective review of patients undergoing laparoscopic surgery and experience hypothermia. The sample size is considerable and the analysis was sound.

Technically, retrospective review should not be called cohort study as those are typically prospective. The definitions otherwise are quite clear and allows repeatability of this study. The incidence of hypothermia was just over a third of the patients which is quite significant. Can the author provide reasons why did happen given it is a major element in the Surgical Care bundle and part of patient setup?

Response: It was a retrospective cohort study since we can determine the cause-and-effect relationship between the exposure and outcomes. In this case, the hypothermia/episode of hypothermia was an exposure, and the effects were the consequences of the hypothermia (postoperative oxygen requirement and longer PACU stay). The term “retrospective review” could not determine the cause-and-effect relationship.

The incidence of hypothermia in our study is quite high at 34.9% but it is still within the normal range found. The reasons for this high incidence could be due to the lower room temperature inside the operating rooms in our institution (18-20˚C), longer duration of surgery resulting in exposing surgical sites for a long time. Our previous protocol was as follow. Forced air warming devices are applied above the patient in every case and use of warm irrigation and warm intravenous fluid. In case the body temperature turns lower than 36 ˚C, the operating room’s temperature could be adjusted to 22-24 ˚C case by case. Yet, there was still not adequately proper warming throughout the operation. Thus, the more proper guideline mentioned in the “Implication of the study” subsection in the discussion section must be implemented (Lines 329-339). 

Can the author provide insights on how this can be prevented and that high percentage be brought down? Would adding it to the checklist help? How does this figures compare to international literature? It would hugely beneficial if the authors provide insights to how it happens and recommendations on how to prevent it.

Response: The literature mentions the incidence of hypothermia in laparoscopic surgery to be around 29-90%. So our incidence of 34.9% still falls within that range. We think this can be prevented, as we mentioned the more proper guideline in the “Implication of the study” subsection in the discussion section must be implemented (Lines 329-339) “At the preoperative period, proper preoperative warming for at least 30 minutes should be considered in high-risk patients as found previously; however, one should be aware that the risk of burns and thermal injuries may increase. At the intraoperative period, continuously monitoring and maintaining core body temperature above 36˚C must be performed by various methods such as forced air warming devices applied under or above the patient, optimal operating room temperature, warm intravenous fluids, limiting the exposure time of surgical sites, and using warmed, humidified insufflation gases. In the PACU period, continuous warming strategies should still be considered if the core temperature remains below 36 ˚C to minimize possible adverse events. All staff should be trained on management protocols and made aware of their importance and the possible complication of aggressive warming.”

Having a checklist could also help, as one will notice the baseline temperature and will be able to perform the active interventions on time if there is hypothermia.

Reviewer #2:

This study explores the impact of intraoperative hypothermia on patients undergoing laparoscopic surgery, which is clinically innovative and of great significance. The research method is a retrospective cohort study with a large and comprehensive sample size, which is reliable. The setting of independent and dependent variables is reasonable and comprehensive, the statistical methods are reasonable, logistic regression and linear regression are used to determine the relationship, and there is also a clear flowchart. But there are also some shortcomings:

1.It is recommended to replace the reference with the latest one as it has a relatively long history.

Response: We add the two more update references (16 and 34).

2.The conclusion section can provide a clearer indication of the specific direction and focus of future research.

Response: We edit the conclusion on the matter (Lines 368-371).

3.In table4, table5, and table6, abbreviations for some variables (such as OR, CI, etc.) should be given their full names and explanations when they first appear to facilitate readers' understanding. For all tables, it is recommended to add more statistical information, such as the specific calculation method of P-value and the confidence level of the confidence interval.

Response: Thank you for the advice. We add the full name of OR (odds ratio)/ β (Beta-coefficient) and CI (confidence interval) in the legend of Table 4-6. The 95% CI, the wald p-value for each subgroup of the variable (Table 4-6) and p-value by the likelihood ratio (LR) test for categorical outcome (Table 4,5) and p-value by F test statistic for continuous outcome (Table 6) were also mentioned in the heading of each table.

We uploaded our figure files to the Preflight Analysis and Conversion Engine (PACE) digital diagnostic tool and already got the PACE corrected files for resubmission. Thank you so much for your advice.

Should you have more comments, please don’t hesitate to contact us

Thank you for your attention to our paper

Sincerely yours,

Dr. Tashi Penjore

Assoc. Prof. Maliwan Oofuvong

Assist. Prof. Sunisa Chatmongkolchart

Assist. Prof. Chanatthee Kitisiripant

Rongrung Rueangchira-urai

Jaranya Leeratiwong

---

## [Decision Letter · Decision Letter 1]

11 Nov 2024

PONE-D-24-29030R1Effects of Intraoperative Hypothermia on Patients Undergoing Laparoscopic Surgery: A Retrospective Cohort StudyPLOS ONE

Dear Dr. Oofuvong,

Thank you for submitting your manuscript to PLOS ONE. After careful consideration, we feel that it has merit but does not fully meet PLOS ONE’s publication criteria as it currently stands. Therefore, we invite you to submit a revised version of the manuscript that addresses the points raised during the review process.

We look forward to receiving your revised manuscript.

Kind regards,

Marco Clementi, Associate Professor

Academic Editor

PLOS ONE

**Journal Requirements:**

**Additional Editor Comments:**

Thank you for your hard work in the review.

Minor revisions will be made in accordance with the reviewer's request.

Reviewers' comments:

Reviewer's Responses to Questions

**Comments to the Author**

1. If the authors have adequately addressed your comments raised in a previous round of review and you feel that this manuscript is now acceptable for publication, you may indicate that here to bypass the “Comments to the Author” section, enter your conflict of interest statement in the “Confidential to Editor” section, and submit your "Accept" recommendation.

Reviewer #2: (No Response)

2. Is the manuscript technically sound, and do the data support the conclusions?

Reviewer #2: Yes

3. Has the statistical analysis been performed appropriately and rigorously? 

Reviewer #2: Yes

4. Have the authors made all data underlying the findings in their manuscript fully available?

Reviewer #2: Yes

5. Is the manuscript presented in an intelligible fashion and written in standard English?

Reviewer #2: Yes

6. Review Comments to the Author

**Reviewer #2:** 1、The introduction mentions that the adverse effects of intraoperative hypothermia during open surgery include surgical site infections, coagulopathy and increased transfusion requirements, increased analgesic needs, altered drug metabolism, and adverse cardiac events. However, in the context of laparoscopic surgery, why is only arrhythmia mentioned among these adverse effects?

2. In this study, various methods are used to monitor body temperature, and there may be errors and inconsistencies in accuracy between different methods. Could this affect the accuracy of the results? Is it necessary to conduct a subgroup analysis? 3. A temperature monitoring interval of every 30 minutes may be too long; this should be discussed as it represents a limitation or an area for improvement in future studies. Continuous monitoring could provide statistics on the duration of hypothermia, thereby increasing the sensitivity of the research.

7. PLOS authors have the option to publish the peer review history of their article (what does this mean?). If published, this will include your full peer review and any attached files.

Reviewer #2: No

---

## [Author Response · Author response to Decision Letter 1]

16 Nov 2024

Dear PLOS ONE Editor,

We would like to resubmit a manuscript entitled “Effects of Intraoperative Hypothermia on Patients Undergoing Laparoscopic Surgery: A Retrospective Cohort Study” as an original article in your journal.

We are thankful to have a chance to revise our manuscript. To address the editor’s comment, we have added more information in the result, Table 1, S1 Table and Table 3 and discussion section. We also highlighted the changes in the current version. Attached is a point-by-point response to the reviewer’s concerns. 

Journal Requirements:

Answer: We already checked all cited references. There is no retracted article since the full articles of all cited references could be accessed. 

Comments to the Author

Reviewer #2: 1、The introduction mentions that the adverse effects of intraoperative hypothermia during open surgery include surgical site infections, coagulopathy and increased transfusion requirements, increased analgesic needs, altered drug metabolism, and adverse cardiac events. However, in the context of laparoscopic surgery, why is only arrhythmia mentioned among these adverse effects?

Answer: The introduction does mention the effects of intraoperative hypothermia in open surgeries as noted, but there was very little literature looking directly at the consequences of hypothermia in laparoscopic surgeries that also had inconsistent findings. Hence, the consequences of hypothermia were shown in Table 3. We also add the outcome of “intraoperative blood transfusion” into Table 3 (Lines 193-194) and S1 Table. Even though the hypothermia group was related to a higher proportion of blood transfusion requirements, it was not statistically significant in the multivariate logistic regression model. Since it is a retrospective cohort study, the surgical site infection and postoperative coagulopathy are quite limited to determine due to unwell documentation postoperatively. However, we measured the outcome that had more impact and could represent the undesirable consequences of hypothermia, such as duration of ICU admission and length of hospital stay.

2. In this study, various methods are used to monitor body temperature, and there may be errors and inconsistencies in accuracy between different methods. Could this affect the accuracy of the results? Is it necessary to conduct a subgroup analysis? 

Answer: It is true that there are various methods used for recording temperature in our study, but the esophageal route, nasopharyngeal, and rectal temperatures all represent the core body temperature, so it should be accurate. In the case of the temperature measured from the ear (tympanic membrane), since it is less accurate with a potential error of less than 0.5 ˚C, we adjusted the recorded temperatures by adding 0.5 ˚C to correct for this discrepancy. We add the methods of temperature monitoring into Table 1. It showed a balance of methods of temperature monitoring among the hypothermia and normothermia groups (p = 0.334); therefore, the subgroup analysis among methods of temperature monitoring is not necessary. Only two patients were recorded by the rectal route, which required combination with other routes to accomplish comparison by the Chi-square test.

3. A temperature monitoring interval of every 30 minutes may be too long; this should be discussed as it represents a limitation or an area for improvement in future studies. Continuous monitoring could provide statistics on the duration of hypothermia, thereby increasing the sensitivity of the research.

Answer: We monitored the temperature continuously, but we only recorded every 30 minutes (S1 Fig). However, the duration of hypothermia was also measured to see the trend of the temperature change to bring more accuracy to the temperature’s data. We add this limitation into “The limitation subsection” (Lines 354-358). Longer duration of hypothermia could result in more undesirable outcomes; however, we performed different methods of warming (force air warming, warming mattress) (Lines 97-98) to raise the body’s temperature up to normal (BT > 36 ˚C). during surgery. Therefore, the development of a second episode of hypothermia after temperature resumed to normal was considered an exposure to hypothermia-related outcomes/consequences of hypothermia in our laparoscopic patients.

Should you have more comments, please don’t hesitate to contact us

Thank you for your attention to our paper

Sincerely yours,

Dr. Tashi Penjore

Assoc. Prof. Maliwan Oofuvong

Assist. Prof. Sunisa Chatmongkolchart

Assist. Prof. Chanatthee Kitisiripant

Rongrung Rueangchira-urai

Jaranya Leeratiwong

---

## [Editor Report · Decision Letter 2]

20 Nov 2024

Effects of Intraoperative Hypothermia on Patients Undergoing Laparoscopic Surgery: A Retrospective Cohort Study

PONE-D-24-29030R2

Dear Dr. Maliwan Oofuvong

We’re pleased to inform you that your manuscript has been judged scientifically suitable for publication and will be formally accepted for publication once it meets all outstanding technical requirements.

Kind regards,

Marco Clementi, Associate Professor

Academic Editor

PLOS ONE

---

## [Editor Report · Acceptance letter]

3 Jan 2025

PONE-D-24-29030R2 

PLOS ONE

Dear Dr. Oofuvong, 

I'm pleased to inform you that your manuscript has been deemed suitable for publication in PLOS ONE. Congratulations! Your manuscript is now being handed over to our production team.

Kind regards, 

on behalf of

Dr. Marco Clementi 

Academic Editor

PLOS ONE